# Achieving Discharge Limits in Single-Stage Domestic Wastewater Treatment by Combining Urban Waste Sources and Phototrophic Mixed Cultures

**DOI:** 10.3390/microorganisms11092324

**Published:** 2023-09-15

**Authors:** Sandra Chacon-Aparicio, John Alexander Villamil, Fernando Martinez, Juan Antonio Melero, Raul Molina, Daniel Puyol

**Affiliations:** Chemical and Environmental Engineering Group, University Rey Juan Carlos, 28933 Madrid, Spain; sandra.chacon@urjc.es (S.C.-A.); fernando.castillejo@urjc.es (F.M.); juan.melero@urjc.es (J.A.M.); raul.molina@urjc.es (R.M.)

**Keywords:** purple phototrophic bacteria, photo-biorefineries, domestic wastewater treatment, resource recovery, CSTR photobioreactors, membrane photobioreactors

## Abstract

This work shows the potential of a new way of co-treatment of domestic wastewater (DWW) and a liquid stream coming from the thermal hydrolysis of the organic fraction of municipal solid waste (OFMSW) mediated by a mixed culture of purple phototrophic bacteria (PPB) capable of assimilating carbon and nutrients from the medium. The biological system is an open single-step process operated under microaerophilic conditions at an oxidative reduction potential (ORP) < 0 mV with a photoperiod of 12/24 h and fed during the light stage only so the results can be extrapolated to outdoor open pond operations by monitoring the ORP. The effluent mostly complies with the discharge values of the Spanish legislation in COD and *p*-values (<125 mg/L; <2 mg/L), respectively, and punctually on values in N (<15 mg/L). Applying an HRT of 3 d and a ratio of 100:7 (COD:N), the presence of PPB in the mixed culture surpassed 50% of 16S rRNA gene copies, removing 78% of COD, 53% of N, and 66% of P. Furthermore, by increasing the HRT to 5 d, removal efficiencies of 83% of COD, 65% of N, and 91% of P were achieved. In addition, the reactors were further operated in a membrane bioreactor, thus separating the HRT from the SRT to increase the specific loading rate. Very satisfactory removal efficiencies were achieved by applying an HRT and SRT of 2.3 and 3 d, respectively: 84% of COD, 49% of N, and 93% of P despite the low presence of PPB due to more oxidative conditions, which step-by-step re-colonized the mixed culture until reaching >20% of 16S rRNA gene copies after 49 d of operation. These results open the door to scaling up the process in open photobioreactors capable of treating urban wastewater and municipal solid waste in a single stage and under microaerophilic conditions by controlling the ORP of the system.

## 1. Introduction

Worldwide, more than 80% of domestic wastewater (DWW) is discharged into rivers or the sea without any treatment to remove pollutants, which is why the United Nations has expressly set the sixth Sustainable Development Goal (SDG) to ensure the availability and sustainable management of water and sanitation for all [1]. At the same time, governments and institutions are focusing on wastewater reuse strategies to recover both water and nutrients. These facts highlight the need to invest in different wastewater treatment, recycling, and reuse technologies, in line with the Circular Economy Action Plan proposed by the EU in 2020 to achieve a fully circular economy in Europe by 2050 [2].

Developing technologies capable of avoiding water scarcity is of great interest from an agri-food viewpoint that can recover wastewater as irrigation water for crops [3,4,5], as it is rich in nitrogen and phosphorus, which are the basis for stimulating plant growth [6]. All of this has promoted institutions to set limitations on discharge values; for example, in Spain, minimum COD (125 mg/L), nitrogen (15 mgN/L), or phosphorus (2 mgP/L) values are established to be able to reuse wastewater [7]. The key technology in wastewater treatment plants (WWTPs) is biological treatment to comply with legislation and recover water resources in compliance with the circular economy concept [8,9,10]. Nowadays, biological treatments focus on activated sludge, but they involve a high energy cost in the aeration and flocculation of the process, consuming up to 45% and 75% of the plant’s energy [11]. Aerobic granular sludge (AGS) reactors have been optimized to use only one SBR-type reactor. Still, the long start-up period to obtain maturated granules (up to months) limited their development [12]. Also, AGS is not usually efficient for complete N removal, and other biological processes are needed. Other studies have shown that wastewater variability does not always ensure COD, N, and P removal, even using dedicated N and P biological removal systems [13,14]. Nevertheless, these systems do not consider resource recovery within a circular economy concept in their design or operation.

Photosynthetic processes focus on using microalgae, consortia between microalgae and bacteria [14], or activated sludge together with microalgae, known as activated microalgae, to improve the removal of nutrients in WWTPs [11]. These processes have good results in removing N and P from wastewater. Still, they cannot treat high amounts of COD, and the current studies usually need average COD inlet values between 178 and 390 mgCOD/L to reach the COD discharge limits [10,15,16]. Thus, some microalgae–bacteria studies needed an external addition of VFA while working in batch mode and still not reaching the COD, N, and P discharge limits depending on the process steps [17]. In addition, the need to incorporate aeration systems made the process more complex and expensive. Although microalgae–bacteria systems achieved discharge values in a single stage, they required aeration of the culture and anaerobiosis stages for optimal selection of the microorganisms, thus increasing their complexity [18].

Photosynthetic biological studies have recently focused on new emerging processes based on purple phototrophic bacteria (PPB) [19,20]. These anoxygenic photosynthetic microorganisms have gained interest due to their metabolic versatility, as they can grow chemotrophically in aerobic conditions and photoheterotrophically by obtaining energy from light in the IR spectrum [21]. Their secondary metabolism allows them to store C in the form of polyhydroxyalkanoates for the production of bioplastics [22,23,24,25,26], to accumulate P in the form of poly-P to be used as fertilizer [27], or to assimilate N to synthesize protein [28,29] of high biological quality due to their amino acids profile [30]. Also, mixed PPB cultures do not need strict anaerobic conditions to sustain the culture but rather a negative oxidation-reduction potential (ORP) [31]. However, dissolved oxygen promotes aerobic bacteria growth, which may outcompete the PPB communities [32]. Maintaining the PPB cultures during wastewater treatment is necessary to achieve whole COD, N, and P recovery, decreasing the CO_2_ emissions and achieving a higher carbon yield due to the assimilative nature of the PPB metabolism [33]. This is challenging and has not been addressed in open reactors treating domestic wastewater. Furthermore, the PPB’s capacity to accumulate organic carbon makes them useful for co-treatment wastewater with organic waste fractions rich in organic fatty acids. In natural sewage, which is an oligotrophic environment with low dissolved organic carbon content, it has been demonstrated that PPB can be dominant by adding an external carbon source (carboxylic acids, sugars, amino acids, or alcohols) to maintain their growth without being outcompeted by microalgae [9,20,29,34]. At least an additional COD between 300 and 400 mgCOD/L is needed to support the development of PPB and reach whole COD, N, and P assimilation in a single-stage process [35]. A breakthrough way to achieve it is through the co-treatment of DWW and a carbon-rich waste source such as the solubilized organic fraction of municipal solid waste (OFMSW), the basis of the Deep Purple concept (https://deep-purple.eu/, accessed on 11 September 2023). The intention to co-treat both waste sources in a single stage, without the need for fermentation of the OFMSW that increases the operational cost of the process, should also be emphasized. The OFMSW has already been individually used to promote the production of PHA using mixed PPB cultures [24]. However, the co-treatment of urban wastewater with the OFMSW using PPB remains unexplored. Variables such as the appropriate COD/N/P ratio, the effect of hydraulic and sludge retention times (HRT and SRT, respectively), and the influence of the ORP should be adequately evaluated to propose this technology as an alternative for treating these wastes.

This study aims to set the treatment conditions for COD, N, and P recovery from domestic wastewater in single-stage open reactors enriched in PPB. The work proposes the co-treatment of DWW and the liquid fraction of thermally hydrolyzed OFMSW in a single stage through monitoring the ORP to explore the influence of the redox potential over the cultures. Reductive conditions are imposed to hinder aerobic bacteria overgrowth and avoid inhibiting the light-harvesting complexes of PPB, preventing them from losing their advantageous photoheterotrophic metabolism. First, the optimum urban waste mix was determined based on the COD/N/P assimilation in batch tests. Then, two continuous stirred-tank reactors (CSTRs) were operated under strict anaerobic and microaerophilic conditions, respectively, and the influence of the ORP was evaluated. This work proposes maintaining a negative ORP instead of imposing severe anaerobic conditions, which decreases the system’s energy costs and simplifies the reactor operation. Finally, two microaerophilic reactors were overloaded by separating the HRT from the SRT through submerged membranes to analyze the effect of the organic overload on the development of microbial populations and the reactors’ performances. Microbial composition along the process was also evaluated to establish the dynamics of the microbial populations inside the reactors. The results from this study will shed light on treating domestic wastewater with PPB in open reactors like open ponds.

## 2. Materials and Methods

### 2.1. Source of Biowastes

This study used two urban wastes: DWW and OFMSW. The OFMSW came from a municipal waste processing plant in Madrid (Spain). Samples were blended, homogenized, and stored in a cold chamber at 4 °C until further use. The OFMSW was thermally hydrolyzed in advance at 150 °C for 38 min, according to previous results [36], being the liquid fraction of the hydrolysate separated by centrifugation and used in this work. The DWW came from the outlet of the primary settlers of two domestic wastewater treatment plants (WWTPs) located in Estiviel, Toledo (Spain), and La Gavia, Madrid (Spain). The biowastes were collected in three different periods. Table 1 shows the characterization of the residues along the various sampling campaigns.

### 2.2. Inoculum

The inoculum for this study was generated in a raceway photobioreactor fed in batch conditions (HRT of 7 days, equal SRT) with a mix between domestic wastewater from Rey Juan Carlos University (Móstoles, Madrid, Spain) and the liquid fraction of the hydrolysate resulting from the thermal hydrolysis of the OFMSW coming from a municipal waste processing plant located in Madrid (Spain). The DWW/OFMSW ratio was not optimized, and a COD:N ratio of 100:5 was maintained, according to previous studies, for promoting PHA accumulation. The raceway was illuminated with solar light and operated during summertime until a stable mixed culture enriched in PPB. Table 1 shows its main physical–chemical characteristics, whereas the Appendix A shows the microbial composition of those genera that appear in percentages higher than 1% in the mixed culture extracted from the raceway.

### 2.3. Experimental Design

#### 2.3.1. Specific Phototrophic Activity (SPA) Tests

The SPA tests served to determine the optimum mix of the DWW from La Gavia with the liquid fraction (LF) of the OFMSW hydrolysate to maximize the growth of purple phototrophic bacteria, obtaining a complete COD, N, and P assimilation from the mixture in a single process. Different DWW/LF mix ratios, resulting in COD:N ratios from 100:7 to 100:3, were tested in 100 mL anaerobic serum vials in a temperature-controlled shaker (at 30 ± 1 °C), continuously shaken at 230 rpm. In all cases, 10 mgVSS/L of the active phototrophic biomass from the anaerobic raceway was used as inoculum. Finally, the headspace of each vial was flushed with argon for 3 min before being sealed with a rubber septum and aluminum crimp seal to ensure anaerobic conditions along the experiment. Two control experiments were also performed, using DWW alone, and the LF dissolved in ultrapure water up to a COD ca. 1600 mgCOD/L. Table 2 shows the initial conditions in each experiment (mixtures and controls) evaluated. The vials were continuously illuminated with IR lamps (Philips, BR125 IR, Spain) at a 15 cm distance, with an average irradiance of 45 ± 5 W/m^2^, and covered with a UV/VIS filtering foil to avoid inlet radiation at wavelength below 700 nm. The spatial distribution of the irradiance in the experimental set-up was determined by taking different measurements along the irradiated surface using a BLUE-Wave spectroradiometer (Stellar Net, Tampa, FL, USA). All SPA experiments were performed in triplicate.

#### 2.3.2. Semicontinuous Photobioreactors

The co-treatment of the mixed urban wastes has been tested in two 2 L cylindrical photobioreactors inoculated with 1% *v*/*v* of a mixed culture of PPB and continuously agitated with a magnetic stirring bar at 300 rpm. Four IR-LED lamps (emitting at 850 nm) illuminated the reactors from two sides for 12 h and switched off after that for the other 12 h, simulating daylight/night periods in the lab. The photobioreactors were operated under two different regimes, CSTR with no biomass retention (said CSTR mode) and CSTR with biomass retention (said membrane bioreactor or MBR mode). In both cases, the photobioreactors were fed in a semicontinuous mode only during the light period, and the temperature and ORP were continuously monitored in each bioreactor.

Table 3 shows the main inlet parameters and the operating conditions during the CSTR and MBR operation. The CSTR mode operation lasted 90 days, and both CSTRs were fed with variable proportions of DWW and LF, with the HRT varied between 3 and 5 days, resulting in four different operational periods (Stages I–IV, Table 3). These conditions were modified dynamically based on the real-time operational performance, trying to achieve discharge limits in terms of COD, N, and P. CSTR-1 was opened to the air under microaerophilic conditions, simulating the environmental conditions of a scaled-up process. CSTR-2 served as an anaerobic control and was hermetically closed to allow the theoretically optimum growth of these bacteria under photoanaerobic conditions.

During the MBR mode operation, which lasted 49 d, the effect of decoupling the HRT and SRT was evaluated, promoting the aging of the biomass to explore its impact on nutrient assimilation. Thus, both reactors operated under microaerophilic conditions, but the SRT was separated from the HRT by using hollow fiber membranes P60 (Zena, Czech Republic) with an average pore size of 0.1 µm and surface area of 0.95 m^2^ to extract the effluent from the photobioreactor. MBR-1 was operated with fixed SRT and HRT (5 and 3.8 days), and the COD/N ratio was changed from 100/6 (Stage V-a) to 100/5 (Stage V-b). In MBR-2, HRT and SRT were step-by-step decreased with a fixed SRT/HRT ratio of 3/4, resulting in different operational stages (VI-a to VI-d), where the COD/N ratio of the first two stages was 100/6, which was reduced to 100/5 in the last ones for a better comparison with MBR-1.

### 2.4. Analytical Methods

Analytical determination of pH, total suspended solids (TSSs) and volatile suspended solids (VSSs), TKN, and COD were carried out following *Standard Methods for the Examination of Water and Wastewater* [37]. The optical density of the mixed cultures was analyzed at 600 nm and 805 nm, as indicative of total bacteria and PPB, respectively, using a VIS-NIR spectrophotometer (V-630, Jasco, Madrid, Spain). NH_4_^+^, PO_4_^3−^, and total phosphorus (TP) were quantified with a Smartchem-140 (Metrohm, Madrid, Spain). Liquid samples were filtered through a cellulose-ester filter of 0.45 μm pore size (Advantech, Japan). The temperature and redox potentials were determined by probe-HI5221, and the dissolved oxygen was measured using probe-HI98193, both from Hanna Instrument (Gipuzkoa, Spain). All measurements were performed in triplicate (SPA experiments) or duplicate (samples from photobioreactor), and the results are shown with the corresponding standard deviations in all cases.

### 2.5. Microbial Community Analysis

Different samples were taken from the biomass inside the photobioreactors during their operation and from the inoculum, DWW, and LF for microbial community analysis. The reactor samples corresponded to the beginning and end of the stages established in Table 3. All of them were stored at −20 °C until DNA extraction.

The DNA extraction of the samples was performed by a FastSpin for Soil Kit (MP-Biomedicals, Santa Ana, CA, USA) according to the manufacturer’s protocol. Details of the method are given in the Appendix A.

## 3. Results and Discussion

### 3.1. Specific Phototrophic Assay (SPA)

Figure 1 shows SCOD, ammonium, and phosphate assimilation during SPA experiments, PPB, and total bacteria growth through the absorbance ratio at 805 nm (peak absorbance related to the light-harvesting complexes of PPB). Results revealed an absence of the development of the characteristics of PPB’s light-harvesting complexes during the experimental time during the treatment of LF or DWW alone. The inhibition of PPB phototrophic activity and growth can be due to the low pH of the LF [38] and low SCOD content in the case of DWW, which is insufficient to provide a sufficient carbon source for the growth of the culture. In both cases, it was accompanied by limited nutrient assimilation (N and P) after 70 h.

The LF/DWW balance is essential to find the optimal COD:N ratio to fully assimilate the medium’s nutrients. Discharge values for N (15 mgN/L) were achieved in the 100:5 and 100:6 tests at 28 h and before 48 h in the 100:3, 100:4, and 100:7 tests. P assimilation was like N except for the 100:3 ratio, leaving a residual P-value of 2.7 mgP/L (Figure 1B), higher than its maximum discharged value (>2 mgP/L).

The efficiency of SCOD assimilation is directly linked to the COD/N ratio. While the initial SCOD increased, the nutrients became more limiting, causing an excess of remnant SCOD at the end of the test. When checking the extremes, the 100:3 test started with the highest SCOD (2080 ± 90 mgSCOD/L), and its low COD:N ratio caused an excessive SCOD remaining in the medium (higher than 800 mg/L). After 48 h, the ammonium was assimilated entirely, but the SCOD was still consumed during the rest of the experimental time. This cannot be due to biomass growth and is suggested to be related to carbon accumulation as PHA in a secondary metabolic pathway that PPB use to attain redox homeostasis [39]. In contrast, the 100:7 ratio test started with the lowest SCOD (1010 ± 60 mgSCOD/L), and therefore, it achieved the lowest concentration by the end of the experiment (180 mgCOD/L). It is well-known that the assimilation of SCOD is directly related to the concentration of nutrients in the medium [40]. Thus, the N and P limitation at low COD/N ratios avoided reaching the 125 mgCOD/L COD discharge limits.

Due to the similarity of the experiments’ COD and nutrient assimilation results (Table 4), the biomass yield was monitored as it is another key performance indicator for maintaining a stable continuous operation. The best values for this parameter were obtained within the intermediate values, notably in the 100:5 test that achieved 0.71 mgVSS/mgCOD. This value rounds around the highest reported in the literature in a previous work (0.72–0.76 mgVSS/mgCOD), where the SCOD came from sugars (fructose and glucose), and the growth was due to the inoculation of pure cultures of *Rsp. rubrum* and *Rb. capsulatus* [28]. Thus, the optimum value for further continuous experiments was set at a COD:N ratio of 100:5, but, in any case, the ratios 100:6 and 100:7 cannot be discarded and can be selected during the reactor’s operation to attain discharge values for COD, N, and P. In short, it has been demonstrated that it is possible to optimize the mix between the DWW and the LF to maximize PPB growth, considerably improving the COD, N, and P assimilation and, in the case of N and P, achieving discharge values in a single step.

### 3.2. Semicontinuous Operation Using CSTR to Evaluate the Anaerobic and Aerobic Conditions: ORP Influence on the Phototrophic Mixed Culture

As the SPA trials demonstrated that the co-treatment of both urban wastes using mixed phototrophic cultures considerably improves the COD, N, and P assimilation, a co-treatment using two CSTRs in semicontinuous mode operation was analyzed. First, the two systems were acclimatized to develop the PPB biomass properly (Stage I). They were operated maintaining a COD:N ratio of 100:5, the one maximizing the biomass yield in the batch experiments. The HRT of 3 d was higher than the minimum time required for the biomass growth reported in Figure 1 and long enough to avoid biomass washout as it corresponds to a dilution rate of 0.33 d^−1^, far below the specific maximum activities for mixed cultures of PPB (1.44–7.2 d^−1^) [33]. After 15 days, an absorbance ratio of 805/600 nm of almost 1.0 and a stable biomass concentration of around 500 mgVSS/L were obtained, providing evidence of a successful start-up of the system.

Figure 2 shows the general performance parameters of the CSTRs during the operational time. In general, the microaerophilic reactor (CSTR-1) performed similarly to the anaerobic one (CSTR-2) in terms of COD, N, and P assimilation. Evenly, the microaerophilic reactor occasionally achieved better results in terms of discharge parameters than the anaerobic system. These results were accompanied by a negative ORP throughout the 90 days of operation, as reported in Figure 3. This fact suggests that a negative ORP fulfills a photobioreactor’s assimilative (non-oxidative) operation requirements. These results contrast with another study that affirms that operating a photobioreactor opened to the air entails an irremediable transition to an aerobic operation that strongly decreases the assimilative behavior of the system [32]. Indeed, negative ORP values avoid inhibiting the photoheterotrophic metabolism and preserve the growth of PPB [41], despite not working under anaerobic and agitated conditions, as suggested previously [25].

On the other hand, as the outdoor operation of the photobioreactors entails periods of light and darkness, the feed must be during the light periods, thus favoring the photoheterotrophic metabolism [23]. In fact, almost 56% of PPB dominated the mixed cultures all over Stages I and II, decreasing to 23% and 16.5% during Stages III and IV, respectively (Figure 4), always being the most representative metabolic group of the consortium in both reactors, above aerobic heterotrophs, strict anaerobes, and facultative aerobes. It is worth mentioning the evident decrease in aerobic microorganisms, such as the Xanthobacteraceae family and genera like *Rhodococcus*, *Halomonas,* or *Luteolibacter,* as the ORPs of the two CSTRs turn into negative values (between −300 and −400 mV) in Stage I, settling in all cases below 9% of the total 16S rRNA gene copies within the culture. The resilience of the PPB community despite the variations in the HRTs and COD:N ratios in the open reactor is also highly relevant.

The HRT had a higher influence on the microbial composition than the COD:N ratios in the open reactor. During Stage I, *Rhodobacter* sp. played a crucial role, where up to 46.5% of the reactor was colonized by this genus, a k-strategist microorganism typical of CSTRs that prevails in low-strength wastewater treatment operations [31]. In this case, when the HRT is 3 days and the COD:N ratio is 100:5, the biomass renewal is higher than in the other posterior stages that operate with a 5-day HRT (III and IV). CSTRs tend to enrich *Rhodopseudomonas* before *Rhodobacter* in discontinuous start-ups, but the light/dark cycles favor the metabolism of *Rhodobacter* over *Rhodopseudomonas* [31], as happened during Stage I. However, *Rhodopseudomonas*, an r-strategist microorganism, appeared later in CSTR-2 as the HRT increased to 5 d (Stage III). The availability of nutrients in the culture seems to cause the relative abundance to be displaced into this genus [35]. *Rhodobacter* appears to have a higher affinity for the nutrients than other PPB genera, as the organic loading rate (OLR) at this stage, 0.41 (0.05) gCOD/L·d, is the highest of all (Figure 4). However, when changing the COD:N ratio to 100:7 during Stage II, e.g., under organic substrate limiting conditions, this genus disappears almost entirely and is replaced by *C39* in CSTR-1 and *Rhodopseudomonas* in CSTR-2. This result contradicts the generally accepted fact that the *Rhodobacter* genus is a k-strategist [30]. Nevertheless, the literature typically considers the growth strategy for the assimilation of organic substrates, not nutrients [42]. *Rhodobacter* may have a lower saturation constant for N and P than *Rhodopseudomonas*, which favored its dominance along Stage I in both reactors and Stage IV in CSTR-2, when the HRT increased, and the nutrients became limiting again when *Rhodobacter* reappeared. In these fluctuations, the dilution rate may also have a role here. Indeed, it has been seen that nitrogen-fixing genera entail changes in microbial evolution where the k-strategists outnumber the r-strategists [43]. However, the results presented here are not conclusive regarding the role of the dilution rate.

Regardless of the conditions of the system, open (CSTR-1) or closed (CSTR-2), working in a 3-day HRT is too short to completely assimilate carbon and nutrients simultaneously from the medium by the mixed PPB culture. Stage I favored attaining discharge values for N and P in both reactors, whereas during Stage II, SCOD discharge values were achieved from 20 d onwards, especially in CSTR-2. During these stages, PPB prevailed over the rest of the microbial community. This may be due to low oxygen concentrations (over-competing aerobes), which were always below 0.5 mg/L, and the generation times of the PPB are shorter than those of the fermentative microorganisms [31]. Thus, PPB are more adaptable because they have less competition regarding substrate consumption. The microaerophilic conditions provided the CSTR-1 system with higher stability and performance by fixing the HRT in 5 days to reach the discharge values for SCOD and P instead of working in anaerobiosis. CSTR-1 during Stages III and IV removed 83% and 85% of SCOD (respectively) and between 86% and 91% of the P fed to the system. CSTR-2 did not exceed 65% N removal in Stages III and IV and failed to reach P discharge values, with a maximum removal rate of 77% in Stage III, which was insufficient. Only CSTR-1 achieved N discharge values between days 50 and 60. These results compare positively with other studies that used SBR in anaerobic conditions. N and P removal was insufficient to reach discharge values, and they were only consumed by 62 ± 2.0% and 51 ± 2.6%, respectively [44]. Thus, strict anaerobic conditions are not the most suitable for nutrient assimilation by the mixed PPB cultures. As PPB maintained their dominancy within the system, microaerophilic operation favored SCOD, N, and P assimilations while maintaining their photoheterotrophic metabolism.

Figure 5 shows the relationship between the COD:N intake and uptake ratios. It compares how efficient the system is in nitrogen assimilation to the influent’s COD:N ratios. The microaerophilic CSTR was more efficient in N assimilation than CSTR-2 in all operative stages, but Stage IV (COD:N uptake ratios are 57% of the intake ratios versus almost 66%, respectively). This result also shows the stability and better performance of the microaerophilic reactor despite the HRT and COD:N conditions, which even allows a higher enrichment of the PPB cultures. Although PPB are microorganisms that have always been reported to have better COD:N removal performance in anaerobic conditions [45], the complexity of the two urban wastes makes these bacteria adapt to the environment by favoring their photoheterotrophic metabolism in periods of light and remaining resilient to other microorganisms thanks to their chemo-organotrophic metabolism in periods of darkness, as long as the COD:N ratio is optimal for their growth. Interestingly, in both cases, the intake-to-uptake COD:N ratios were better at lower HRTs, accompanied by an increased abundance of non-PPB biomass when that efficiency decreased. The rise in anaerobic chemoheterotrophic microorganisms due to longer SRTs may displace the PPB and negatively affect the reactor’s nitrogen uptake efficiency. This suggestion has been further explored by using MBRs.

### 3.3. Semicontinuous Operation Using MBR to Evaluate the SRT and HRT Parameters

Upon the finalization of the CSTR operation, the biomass from CSTR-1 was used to inoculate two microaerophilic MBRs, filling the reactors with half of the CSTR-1 volume and the other half with DWW. Subsequently, the PPB mixed culture was adapted to the new operating conditions for one week by feeding with the mixture of DWW and LF (see Table 3).

The HRT and SRT variations strongly impacted both reactors’ C, N, and P removals. In MBR-1, average COD and P removal values were 86% and 77%, respectively, and on multiple occasions during Stage V-a (day 112–118), 96–100% removal allowed for the reaching of the discharge values in SCOD and P. In MBR-2, the best performance was achieved in Stage VI-a, reaching COD and P removal efficiencies of 87% and 55%, respectively. Despite the low abundance of PPB in both stages (V-a and VI-a), the system operated within the legal limits for COD and P. However, neither reactor could remove enough N to achieve the discharge values of 15 mgN/L during these stages. An N removal of up to 56%, 42%, and 47% (Stage V-a, Stage VI-a, and Stage VI-b, respectively) was insufficient. This fact may be due to the high N concentration in the influent (Figure 6) and the low amount of assimilative microorganisms, like PPB, as will be further discussed.

In the following stages (V-b in MBR-1 and VI-c and VI-d in MBR-2), the COD:N ratio was decreased to 100:5, thus increasing the OLR to 0.3 gCOD/L·d in Stage V-b and from 0.4 to 0.76 gCOD/L·d in Stages VI-c and VI-d, respectively. Both MBR-1 and MBR-2 were able to remove a higher N percentage. MBR-2 achieved N values between 29.9 (0.7) and 24.4 (0.65) mgN/L in Stages VI-c and VI-d, respectively. However, MBR-1 at Stage V-b reached values below the discharge limits (11.2 (0.2) mgN/L) or very close to 15 mgN/L at several points (days 126 to 133). The increase in the N assimilation was accompanied by the proliferation of PPB in the MBR systems, especially in MBR-1, where at the end of Stage V-b, around 22% of the microbial community was composed of PPB taxa such as genus *Rhodobacter* and family *Rhodospirillales* (Figure 7). Similar results have been reported working with DWW in the literature [46] but working under anaerobic conditions and continuous irradiation, which notably increases the process’s costs during scaling up, unlike the operation in the present study.

Due to the negative ORP favored inside the reactors, aerobic microorganisms had a limited proliferation in both systems, selecting anaerobic and facultative cultures (Figure 7). As shown in Figure 8, the ORP between days 100 and 111 turned into positive values in MBR-2, accompanied by increased dissolved oxygen for several days, rising to 1–2 mgO_2_/L. These conditions favored the appearance of aerobic genera up to 22% (Figure 7). During the rest of the stages, the ORP remained negative, and the percentages of aerobes in both reactors never exceeded 10%. This presence of aerobic microorganisms might be promoted by the oxygenic condition of the influents and the contact of the reactors’ surfaces with atmospheric oxygen. Bacteria such as *Thauera* [47] and other aerobes took advantage of the positive ORP to use the organic acids derived from the activity of the fermentative microorganisms as a substrate for their chemo-organotrophic metabolism.

The uncoupling of SRT and HRT substantially impacted the microbial population dynamics. PPB were not able at any point to outcompete anaerobic chemoheterotrophic bacteria (Figure 7) and were strongly displaced by anaerobic chemoheterotrophs at the beginning of the operation in both reactors. Given their fermentative metabolism to generate VFA [30], these taxa strongly impacted the PPB development, so understanding their coexistence is critical to improving and developing robust wastewater treatment systems. This fact was not a detriment to PPB development, as the fermentative metabolism promotes PPB redox homeostasis by sugar fermentation and conversion into VFA that PPB can assimilate easier than sugars [48]. The recovery of PPB populations started during the last operative stages in both reactors, achieving up to 25% and 12% of total 16S rRNA gene copies in MBR-1 and MBR-2, respectively.

The COD:N uptake ratio is another key factor that indicates the presence of assimilative bacteria as PPB in the culture, as shown previously in Figure 5 for the semicontinuous CSTR. The same analysis was conducted for the MBRs, depicted in Figure 9. Working at a COD:N ratio of 100:5, as in Stage V-b and Stages VI-b-c-d, 22% of relative abundance of PPB in MBR-1 was reached. In this reactor, the fermentative genus of *Bacteroides* [49] became highly relevant during Stage V-a, operating at a COD:N ratio of 100:6, but disappeared in Stage V-b when the ratio was changed to 100:5. Similar results are observed in MBR-2. Working back to the 100:5 ratio, the relative abundance of PPB increased and became competitive against fermentative genera and aerobes. The loss of aerobic microorganisms (Stage V-b and Stage VI-d) may be related to the recovery of the PPB population, supporting the possibility that their coexistence is not feasible because they compete for the same substrate. Therefore, it seems clear that aerobic or microaerophilic conditions with positive ORPs are inadequate for developing a PPB community in open reactors, especially under high organic load.

## 4. Implications and Future Perspectives

Meeting the discharge limits set by Spanish legislation for COD, N, and P through the operation of CSTRs with a 3-day HRT in a one-step process, achieving up to 60% relative abundance of PPB, represents a promising outcome for treating domestic wastewater using mixed PPB cultures. This achievement stands out compared to the current best available technology, the activated sludge process, which releases resources like C as CO_2_ and N as N_2_, incurring high energy costs due to aeration processes [50]. Consequently, the emerging PPB technology appears attractive to the industry and society due to its ability to fully recover resources. Nonetheless, this technology has limitations primarily related to light availability, imposing a maximum reactor depth of around 30 cm [51]. This limitation renders it unfeasible to treat urban wastewater in large cities with high surface area requirements. However, it is an attractive decentralized alternative for rural areas and small towns with more surface availability.

In this study, successfully utilizing two urban waste streams in a single stage is crucial to enhancing the production of high-value compounds. This capability is a vital strength of these bacteria. Their versatile metabolism enables them to accumulate PHA, glycogen, pigments, or bacterial protein, making them more market-friendly and aligning with a circular economy. This is a significant stride towards the EU’s goal of achieving climate neutrality by 2050 [1].

However, some parameters, like OLR (Organic Loading Rate) and the availability of light in the reactor, need further optimization to effectively treat effluent while complying with regulations and obtaining biomass rich in high-value products. Several research groups are currently focused on upscaling the process and exploring various reactor types for wastewater treatment. Additionally, they are working on optimizing the extraction of high-value products and considering environmental factors. The core idea behind this technology is that PPB can grow using sunlight, reducing system costs and enhancing their competitiveness in the market.

## 5. Conclusions

This study has demonstrated the possibility of co-treatment of urban wastewater with the liquid fraction of OFMSW in a single step using PPB in 12/12 h light/dark systems, which lowers the economic costs of the process. Stable discharge values in SCOD and P have been consistently achieved by enrichment of up to 50–60% PPB in CSTRs with a 3-day HRT and in microaerophilic conditions with negative ORP and systems with membrane retention of an HRT of 3 d and SRT of 4 d. In these processes, the COD:N ratio should be within the range of 100:4.37–6.63, thus ensuring the removal of C and P. The increase in the SRT by using an MBR improves the quality of the effluent, complying with the regulations for the water reuse in unprotected areas, but causes the anaerobic chemoheterotrophs to prevail in the system, especially at high COD:N ratios (around 100:6). Achieving N discharge limits is challenging but is feasible with strict control of the SRT and the ORP. These results strongly support open anaerobic systems (e.g., open raceways), thus reducing operating costs and making a single-step domestic wastewater treatment that reaches discharge limits feasible.

## Figures and Tables

**Figure 1 microorganisms-11-02324-f001:**
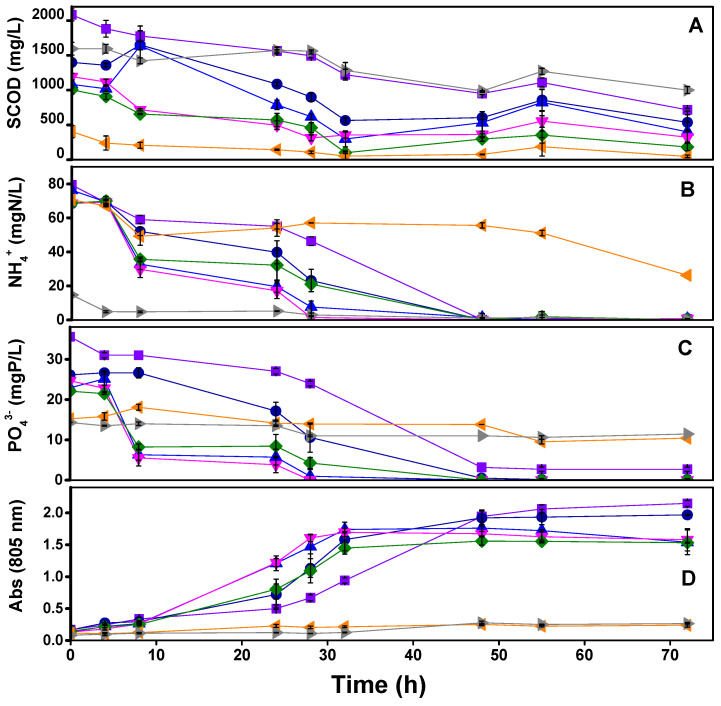
Time course of SCOD (**A**), ammonium (**B**), phosphate (**C**), and absorbance at 805 nm (**D**) in the SPA experiment with SCOD:N ratios 100:3 (-■-), 100:4 (-●-), 100:5 (-▲-), 100:6 (-▼-), 100:7 (-◆-), control of DWW (-◀-), and control of LF (-▶-). Error bars are standard deviations from triplicate measurements.

**Figure 2 microorganisms-11-02324-f002:**
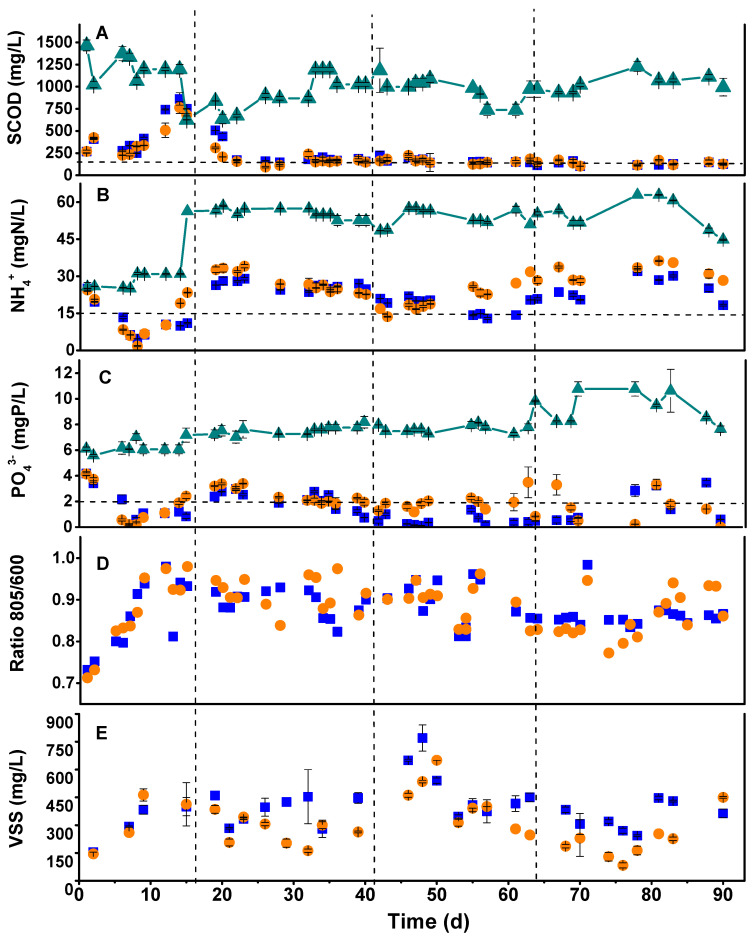
Time course of SCOD (**A**), ammonium (**B**), phosphate (**C**), 805 to 660 nm absorbance ratio of the biomass (**C**,**D**), and volatile suspended solids (**E**) in the effluent of the microaerophilic CSTR-1 (■), the anaerobic CSTR-2 (●), and the influent of both reactors (-▲-). The horizontal dotted lines in panels A, B, and C show the limits of discharge values according to the Spanish legislation. The operative stages are shown on the graph (from I to IV).

**Figure 3 microorganisms-11-02324-f003:**
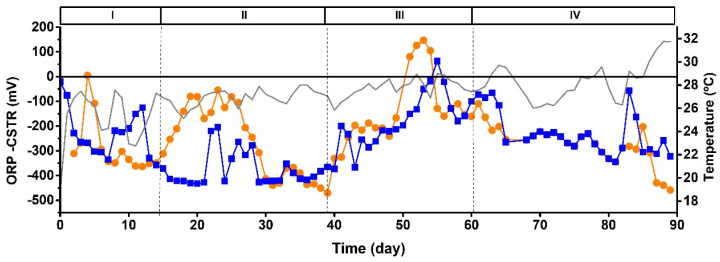
Time course of ORP in CSTR-1 microaerophilic (**-■-**) and CSTR-2 anaerobic (**-●-**) and room temperature (--). The operative stages are shown on the graph (from I to IV).

**Figure 4 microorganisms-11-02324-f004:**
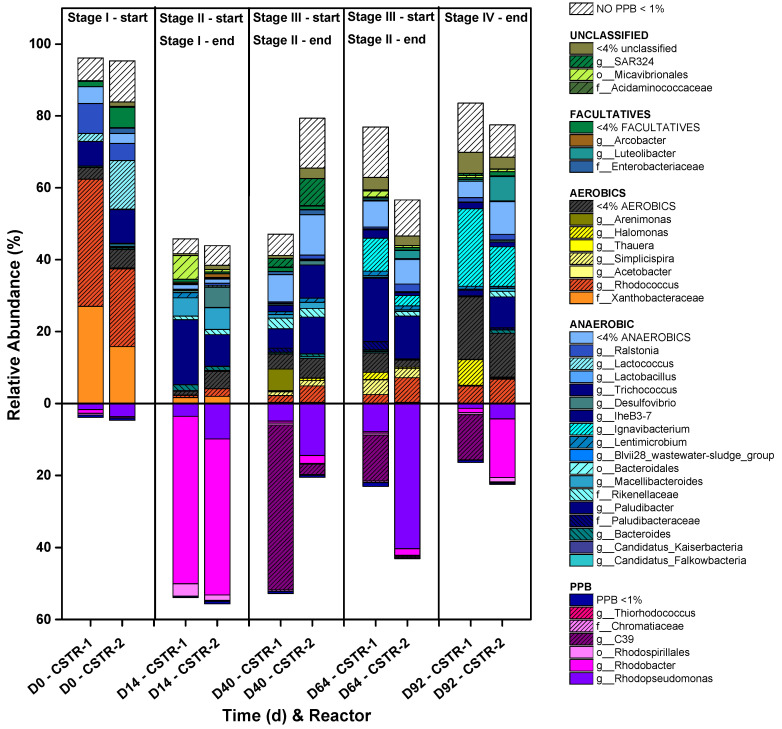
Evolution of the microbial community of the CSTR mode operation during the different operative stages. Values over zero represent non-PPB microorganisms, whereas values below zero show PPB microorganisms. Only taxa accounting for more than 4% of the 16S rRNA gene copies are identified, whereas the rest are summed up in a single identifier. The Appendix A also shows the taxa that appear in percentages from 1 to 4%.

**Figure 5 microorganisms-11-02324-f005:**
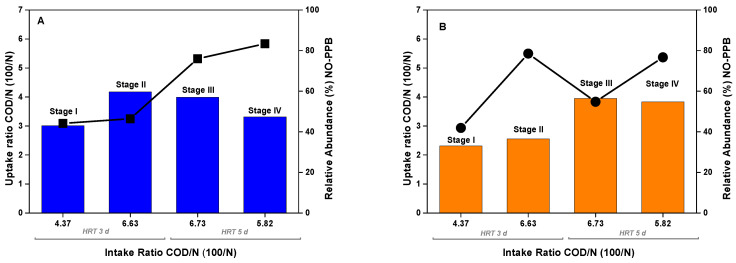
Relation between COD:N intake and uptake ratios (bars) in each stage compared with the relative abundance of non-PPB microorganisms (symbols), expressed as 16S rRNA gene copies. Graph (**A**) represents CSTR-1, and Graph (**B**) represents CSTR-2.

**Figure 6 microorganisms-11-02324-f006:**
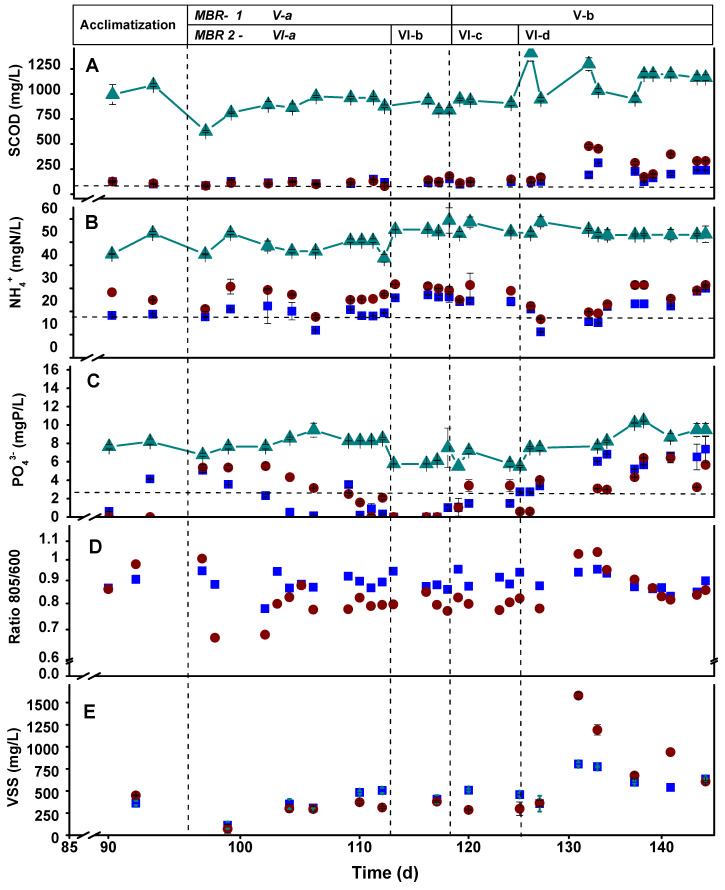
Time course of SCOD (**A**), ammonium (**B**), phosphate (**C**), growth (**CD**), and volatile solid suspension (**E**) in MBR-1 (■) and MBR-2 (●) and influent (-▲-) in both reactors. The horizontal dotted lines in graphs A, B, and C show the limits of discharge values in Spanish legislation. The different stages (from V to VI) are at the top of the figure.

**Figure 7 microorganisms-11-02324-f007:**
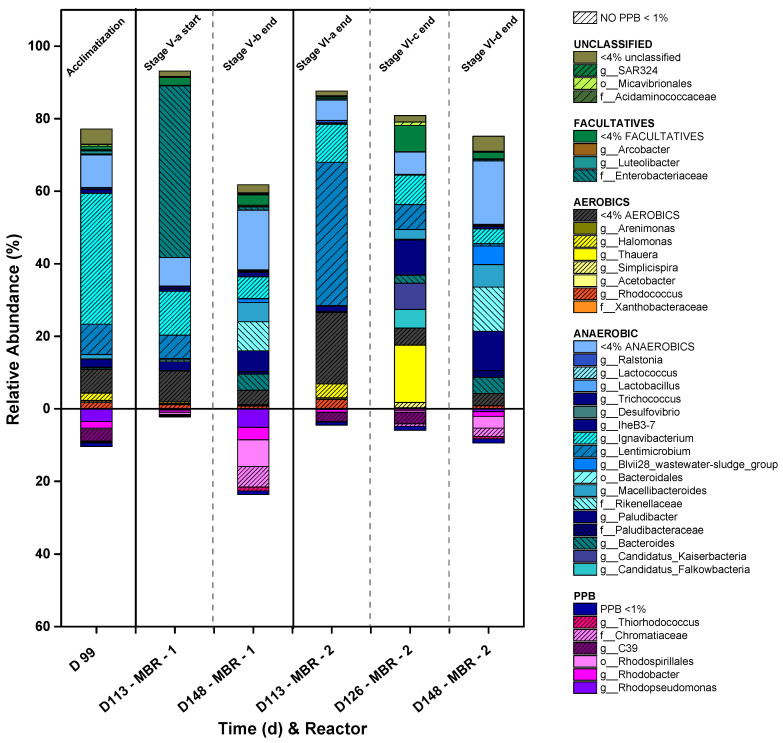
Evolution of the microbial community of the MBR mode operation during the different stages. (The Appendix A shows the evolution of the microbial composition in more detail, including the genera that appear in percentages higher than 1%).

**Figure 8 microorganisms-11-02324-f008:**
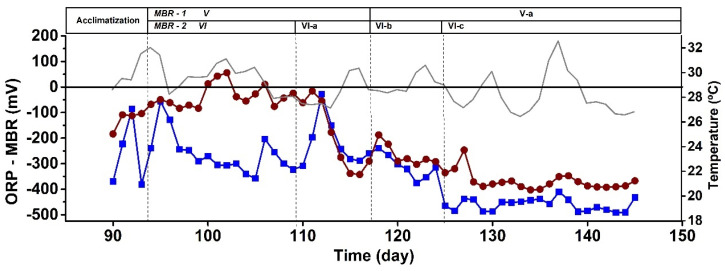
Time course of ORP in MBR-1 (**-■-**) and MBR-2 (**-●-**) and temperature (--) in both reactors. The different stages (from V to VI) are at the top of the figure.

**Figure 9 microorganisms-11-02324-f009:**
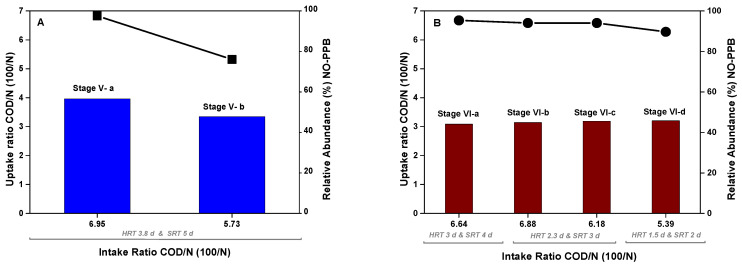
Relation between COD:N intake and uptake ratios (bars) in each stage compared with the relative abundance of non-PPB microorganisms (symbols) expressed as 16S rRNA gene copies. Graph (**A**) represents MBR-1, and graph (**B**) represents MBR-2.

**Table 1 microorganisms-11-02324-t001:** Average values with standard deviations (in brackets) for the soluble COD, ammonium (NH_4_^+^), and phosphate (PO_4_^3−^) and volatile suspended solids (VSSs) liquid fraction of the hydrolysate from the OFMSW, DWW from Estiviel, Toledo (Spain), DWW from La Gavia, Madrid (Spain), and inoculum of PPB.

	SCOD (mg/L)	TCOD (mg/L)	NH_4_^+^ (mgN/L)	PO_4_^3−^ (mgP/L)	VSS (mg/L) *
Campaign 1: Days 1–39					
OFMSW hydrolysate (n = 9)	67,000 (1000)	78,000 (3000)	59 (1)	75 (4)	58 (1)
DWW Estiviel (n = 18)	196 (6)	270 (8)	48 (1)	5 (1)	26 (4)
Campaign 2: Days 40–60					
OFMSW hydrolysate (n = 9)	62,000 (2000)	78,000 (3000)	37 (2)	132 (2)	58 (1)
DWW Estiviel (n = 18)	168 (3)	270 (8)	54 (1)	6 (1)	24 (2)
Campaign 3: Days 61–147					
OFMSW hydrolysate (n = 9)	51,000 (1000)	57,000 (300)	85 (4)	130 (20)	53 (1)
DWW Estiviel (n = 18)	106 (3)	500 (22)	48 (2)	5 (1)	90 (15)
SPA source biowastes					
OFMSW hydrolysate (n = 9)	67,000 (1000)	78,000 (3000)	59 (1)	75 (4)	58 (1)
DWW La Gavia (n = 4)	155 (10)	203 (20)	65 (1)	16 (1)	69 (10)
Inoculum (SPA and semicontinuous photobioreactors)					
Inoculum of PPB (n = 3)	330 (10)	800 (20)	6 (1)	5 (1)	368 (3)

* g/L in case of OFMSW.

**Table 2 microorganisms-11-02324-t002:** Average and 95% confidence intervals from triplicates of measured macroscopic values of the five mixtures of LF and DWW and the two negative control experiments (LF and DWW).

COD:N Ratio	SCOD (mg/L)	NH_4_^+^ (mgN/L)	PO_4_^3^ (mgP/L)	VSS (mg/L)	TOC (mg/L)	pH
100:3	2080 ± 90	62 ± 1	11.6 ± 0.6	92.0 ± 0.0	890.5 ± 59.4	6.92 ± 0.02
100:4	1400 ± 40	53 ±1	8.5 ± 0.4	100 ± 10	587.2 ± 39.1	6.91 ± 0.02
100:5	1080 ± 50	59 ± 2	8 ± 2	104 ± 5	554.8 ± 37.0	6.9 ± 0.02
100:6	1190 ± 50	53 ± 1	8.0 ± 0.7	92 ± 5	478.8 ± 31.9	7.03 ± 0.02
100:7	1010 ± 60	53 ± 3	7.2 ± 0.2	88 ± 5	401.3 ± 26.8	7.04 ± 0.02
DWW (control)	400 ± 90	54.8 ± 0.6	5.0 ± 0.9	86 ± 8	54.4 ± 3.6	7.55 ± 0.02
LF (control)	1600 ± 90	11.4 ± 0.6	4.7 ± 0.1	41 ± 2	716.8 ± 47.8	4.46 ± 0.02

**Table 3 microorganisms-11-02324-t003:** CSTR and MBR mode operating conditions and inlet feed parameters during the trial. The average of the feed input values is represented with standard deviations (in brackets). The error in COD to N ratios is estimated with the 95% confidence intervals.

	CSTR Mode								MBR Mode					
Reactor	CSTR-1 and CSTR-2								MBR-1		MBR-2			
Stage	I		II		III		IV		V-a	V-b	VI-a	VI-b	VI-c	VI-d
Period (d)	01–15		15–40		40–62		62–96		96–118	118–145	96–111	111–118	118–125	125–145
Time (d)	15		25		22		34		22	27	15	7	7	20
HRT (d)	3		3		5		5		3.8	3.8	3	2.3	2.3	1.5
SRT (d)	3		3		5		5		5	5	4	3	3	2
COD:N	100:4.4 ± 1.7		100:6.6 ± 0.9		100:6.7 ± 0.7		100:5.8 ± 0.6		100:6.9 ± 0.6	100:5.8 ± 0.6	100:6.6 ± 0.8	100:6.9 ± 0.7	100:6.2 ± 1.4	100:5.4 ± 0.6
OLR (g COD/L·d)	0.41 (0.05)		0.30 (0.07)		0.21 (0.04)		0.21 (0.05)		0.23 (0.03)	0.30 (0.04)	0.29 (0.05)	0.39 (0.02)	0.40 (0.01)	0.76 (0.09)
SLR (g COD/gVSS·d) *	1.6 (0.6)	1.6 (0.7)	0.8 (0.3)	1.0 (0.3)	0.40 (0.09)	0.49 (0.15)	0.6 (0.2)	1.0 (0.3)	0.8 (0.5)	0.6 (0.2)	1.0 (0.1)	1.17 (0.07)	1.2 (0.3)	1.2 (0.6)
Inlet parameters (mg/L)														
SCOD	1200 (40)		970 (20)		1010 (40)		1100 (40)		890 (20)	1100 (30)	890 (20)	870 (30)	910 (20)	1210 (30)
NH_4_^+^-N	32 (1)		56 (1)		54 (1)		55 (1)		51 (1)	55 (2)	50 (1)	54 (2)	57 (2)	54 (2)
PO_4_^3−^-P	6 (1)		7 (1)		8 (1)		9 (1)		8 (1)	8 (1)	8 (1)	7 (1)	7 (1)	9 (1)

* In CSTR mode, R1 and R2 data are split.

**Table 4 microorganisms-11-02324-t004:** Resume of the five sets of COD:N ratios (100:3 to 100:7) and the controls for the SPA. Calculation of COD:N:P intake, removal rates, biomass yield, and pH variation during the SPA.

Ratio Test	COD:N: P Intake			Removal (%)			pH		*Y _X_*_/*S*_ (*mgVSS*/*mgCOD*)
	COD	N	P	COD	N-NH_4_^+^	P-PO_4_^3−^	Initial	Final	
100:3	100	5.1	0.9	64.1	99.4	91.9	6.9 ± 0.1	6.9 ± 0.1	0.48 ± 0.01
100:4	100	7.4	1.2	58.6	98.8	100.0	6.9 ± 0.1	7.0 ± 0.1	0.68 ± 0.10
100:5	100	10.9	1.4	59.6	99.1	100.0	6.9 ± 0.1	6.9 ± 0.1	0.71 ± 0.03
100:6	100	6.4	1.0	74.2	98.8	100.0	7.0 ± 0.1	6.8 ± 0.0	0.46 ± 0.06
100:7	100	6.8	0.9	84.5	99.9	100.0	7.0 ± 0.1	6.9 ± 0.1	0.27 ± 0.09
DWW	100	10.9	0.7	86.8	62.9	41.7	7.5 ± 0.1	7.8 ± 0.1	0.26 ± 0.05
LF	100	1.4	0.3	30.6	52.4	28.7	4.5 ± 0.1	4.8 ± 0.1	0.21 ± 0.05

## Data Availability

All datasets of this paper are available upon reasonable request.

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
