# Peer review of "Achieving Discharge Limits in Single-Stage Domestic Wastewater Treatment by Combining Urban Waste Sources and Phototrophic Mixed Cultures"

_microorganisms, 2023, doi:10.3390/microorganisms11092324_

Round 1

Reviewer 1 Report

Manuscript ID : microorganisms-2557098

My evaluation of the article led me to the conclusion that it raises some issues that should be addressed before publishing. The study's concept is sound and new to the field. I give revisions for the article's current format in order to bring it up.  

General:

1.     The title and keywords are good.

2.     The abstract seems to be perfect and precisely stated. Please review the English errors in this section.

3.     What is novelty of the review?

4.     Author should highlight the research gap.

5.     Can you precisely highlight the novelty of the study, so the reader will not be confused?

6.     References 3-5 are wrongly cited. Please follow the journal guidelines. Please revise your paper accordingly since some issues occur in several spots in the paper. It is not important to cite too many articles. Please remove 3-5 and simply cite this single reference here- Guerrero–Barajas, C.; Ahmad, A.; Ibrahim, M.N.M.; Alshammari, M.B. Advanced Technologies for Wastewater Treatment. In Green Chemistry for Sustainable Water Purification; Shahid-ul-Islam, Shalla, A.H., Shahadat, M., Eds.; Wiley: Hoboken, NJ, USA, 2023; pp. 179–202. ISBN 978-1-119-85229-2. 

7.     The whole manuscript, especially the introduction, also needs a sincere English revision.

8.     The main objective of the work must be written on the more clear and more concise way at the end of introduction section.

9.     Please provide space between number and units. Please revise your paper accordingly since some issue occurs on several spots in the paper.

10.  The quality of figure 1 is very low.

11.  The results are simply demonstrated, but there is no comparison with the literature or scientific comment. Why?

12.  Please add a section about challenges and future perspectives.

13.   Please add a comparative profile section. The data is not enough so please add these two section to improve the standard to meet the journal standard.

14.  What about the replication of data?

15.  Conclusion section is missing some perspective related to the future research work, quantify main research findings, and highlight relevance of the work with respect to the field aspect.

16.  To avoid grammar and linguistic mistakes, MAJOR level English language should be thoroughly checked. Please revise your paper accordingly since several language issue occurs on several spots in the paper.

17.  Reference formatting need carefully revision. All must be consistent in one format. Please follow the journal guidelines.

Recommendation: Major REVISION

The whole manuscript, especially the introduction, also needs a sincere English revision. Extensive editing of English language required 

Author Response

REVIEWER 1.

  1. The title and keywords are good.

Thanks for the comment.

  1. The abstract seems to be perfect and precisely stated. Please review the English errors in this section.

Thanks for the remark. The English of the entire paper has been carefully checked using the Premium version of the Grammarly software. We have also checked the consistency of the statements to clarify the text.

  1. What is novelty of the review? AND 4.Author should highlight the research gap.

Thanks for the comment. This research demonstrates that it is possible to recover C, N, and P from domestic wastewater by using PPB mixed cultures in a single-stage process and open to the atmosphere. This has been never published before, as either it has been necessary to work under anaerobic conditions or to add an external and expensive source of organic carbon to attain the discharge limits and achieve full C, N, and P recovery, or even both. The novelty of this research is in the ability of PPBs to recover C, N, and P from the co-treatment of wastewater and domestic organic waste under microaerophilic conditions, achieving the same results, or even better, as anaerobic conditions. The scaling up of the process requires working in open conditions to reduce operational costs.

  1. Can you precisely highlight the novelty of the study, so the reader will not be confused?

Thanks for the comment. The paper's objective has been modified to make clear the novelty and the reason for the research. The following text has been added in the introduction:

However, dissolved oxygen promotes aerobic bacteria growth, which may outcompete the PPB communities [33]. Maintaining the PPB cultures during the wastewater treatment is necessary to achieve whole COD, N, and P recovery, decreasing the CO2 emissions and achieving a higher Carbon yield due to the assimilative nature of the PPB metabolism [34]. This is challenging and has not been addressed in open reactors treating domestic wastewater.

And the objectives paragraph has been corrected as follows:

This study aims to set the treatment conditions for COD, N, and P recovery from domestic wastewater in single-stage open reactors enriched in PPB. The work proposes the co-treatment of DWW and the liquid fraction of thermally hydrolyzed OFMSW in a single stage through monitoring the ORP to explore the influence of the redox potential over the cultures. Reductive conditions are imposed to hinder aerobic bacteria overgrowth and avoid inhibiting the light-harvesting complexes of PPB, preventing them from losing their advantageous photoheterotrophic metabolism. First, the optimum urban waste mix was determined based on the COD/N/P assimilation in batch tests. Then, two continuous stirred-tank reactors (CSTR) were operated under strict anaerobic and microaerophilic conditions, respectively, and the influence of the ORP was evaluated. This work proposes maintaining a negative ORP instead of imposing severe anaerobic conditions, which decreases the system's energy costs and simplifies the reactor operation. Finally, two microaerophilic reactors were overloaded by separating the HRT from the SRT through submerged membranes to analyze the effect of the organic overload on the development of microbial populations and the reactors' performance. Microbial composition along the process was also evaluated to establish the dynamics of the microbial populations inside the reactors. The results from this study will shed light on treating domestic wastewater by PPB in open reactors like open ponds.

  1. References 3-5 are wrongly cited. Please follow the journal guidelines. Please revise your paper accordingly since some issues occur in several spots in the paper. It is not important to cite too many articles. Please remove 3-5 and simply cite this single reference here- Guerrero–Barajas, C.; Ahmad, A.; Ibrahim, M.N.M.; Alshammari, M.B. Advanced Technologies for Wastewater Treatment. In Green Chemistry for Sustainable Water Purification; Shahid-ul-Islam, Shalla, A.H., Shahadat, M., Eds.; Wiley: Hoboken, NJ, USA, 2023; pp. 179–202. ISBN 978-1-119-85229-2.

Thanks for the review. We have used the Mendeley Cite with the style for the journal Microorganisms, but we found that the software commits mistakes sometimes. So, we have carefully checked the journal styles and corrected the references list. With regards to the reference suggested by the reviewer, we humbly consider that the references provided are more accurate than the reference suggested by the reviewer, and we have decided to keep the references that we had initially placed as they are more in line with the objective of the text. The reference suggested came from a Book that is not easily accessible unless you pay for it. In any case, we acknowledge the reviewer for the suggestion and we will keep this book in mind for future research.

  1. The whole manuscript, especially the introduction, also needs a sincere English revision.

Thanks for the comment. This suggestion has been addressed in point 2.

  1. The main objective of the work must be written on the more clear and more concise way at the end of introduction section.

Thanks for the review. The objective of the paper has been modified in an attempt to make clear the novelty and the reason for the research. The reviewer can check the response to comment 5.

  1. Please provide space between number and units. Please revise your paper accordingly since some issue occurs on several spots in the paper.

Thanks for the comment. It has been corrected and remarked in the text.

  1. The quality of figure 1 is very low.

Thanks for the comment. The quality of the figures in the Word document is sometimes lower than the file uploaded to the submission system due to the conversion when pasting in the Word file. But all the Figures are in TIFF format and comply with the journal requirements.

  1. The results are simply demonstrated, but there is no comparison with the literature or scientific comment. Why?

Thanks for the comment. We humbly disagree with the comment of the reviewer. There is compare and contrast in the manuscript in most of the paragraphs of the results and conclusion section. However, the existing literature is still limited. In some cases, we have compared with literature on generic domestic wastewater treatment by bioprocesses (PPB, mixed cultures, microalgae, activated sludge, granular sludge, among others). Examples of compare and contrast can be found in Lines 269, 288, 300, 302, 332, 341, 342, 348, 374, 431, 446 and 460.

  1. Please add a section about challenges and future perspectives.

Thanks for the review. We have added, before the Conclusions section, a section in the text entitled ‘Implications and future perspective’.

  1. Please add a comparative profile section. The data is not enough so please add these two section to improve the standard to meet the journal standard.

Thanks for the review. The present research is not a qualitative article, but a quantitative one. The different experiments carried out determine that the proposed objective for wastewater treatment by PPB under microaerophilic conditions is an effective and efficient technology as the anaerobic conditions using PPB. It is even a point of improvement for process scale-up and cost reduction. The length of the paper is now quite large, and we consider that a comparative profile section would not cause enough improvement of the text to be worth it.

  1. What about the replication of data?

Thanks for the comment. In all cases, the samples are replicated, and the standard deviations of each one are calculated and shown in tables and figures. In most cases, the data variability analysis has been conducted by calculating the 95% Confidence Intervals.

  1. Conclusion section is missing some perspective related to the future research work, quantify main research findings, and highlight relevance of the work with respect to the field aspect.

Thanks for the comment. We have added, before the Conclusions section, a section in the text entitled Implications and Future Perspective.

  1. To avoid grammar and linguistic mistakes, MAJOR level English language should be thoroughly checked. Please revise your paper accordingly since several language issue occurs on several spots in the paper.

Thanks for the comment. The reviewer can check our response to this matter in query 2.

  1. Reference formatting need carefully revision. All must be consistent in one format. Please follow the journal guidelines.

Thanks for the suggestion. All the bibliography has been revised again and unified to adjust to the journal requirements.

Reviewer 2 Report

The paper is interesting, although some other tests in the same topic will be necessary. In any case, in the opinion of this reviewer, the main problem is about the references: several names are too often the same in the different cited papers (eg, ref. n. 9, 18, 20, 22, 23, 24, 25, 29, 32, 33, 34, 36, ...). It could also be ok if necessary, but this reviewer thinks if some of this paper is replaced by some others dealing the same argument from different authors, the independence of the paper would be clearer.

Other suggestions are list as below:

Abstract: try to reduce it focusing only on the more significant results (the rest is in the main text).

Line 47-48: in the text the authors refer to Europe limitation, while the cited link refers to a Spanish Decret; please, correct whit an Europeen reference.

Line 56-58: the affermation is important, please, insert some other references, the only n.13 is not enough.

Line 84-85: please, give an explanation for which "wastewater is an oligotrophic environmnet".

Line 97: SRT is conventionally "SLUDGE retention time", although it can be used generally for cells.

Line 127: why do not couple TSS to VSS?

Table 1: this is only a facultative suggestion: it could be preferable to use "sCOD" and "COD" rather than "SCOD" and "TCOD"; more, there is a * (star)  after the measure unit of VSS without an explanation.

Line 297: there is a typing mistake, "Figure 1" should be replaced by "Figure 2"

References: please, prefer international references than national when it is possible, and verify and update the access of links.

Author Response

REVIEWER 2.

The paper is interesting, although some other tests in the same topic will be necessary. In any case, in the opinion of this reviewer, the main problem is about the references: several names are too often the same in the different cited papers (eg, ref. n. 9, 18, 20, 22, 23, 24, 25, 29, 32, 33, 34, 36, ...). It could also be ok if necessary, but this reviewer thinks if some of this paper is replaced by some others dealing the same argument from different authors, the independence of the paper would be clearer.

Thanks for the suggestion. We will keep this in mind for future research. However, in this field, the technology is still maturing and although it is becoming more and more interesting, the research groups dedicated to it are not as many as in more mature technologies (e.g., anaerobic digestion). The references included correspond to the research groups that have the technology more advanced and can be considered the current references for the technology worldwide.

Abstract: try to reduce it focusing only on the more significant results (the rest is in the main text).

Thanks for the comment. We have reduced the content of the abstract, trying not to omit relevant research information.

Line 47-48: in the text the authors refer to Europe limitation, while the cited link refers to a Spanish Decret; please, correct whit an Europeen reference.

Thanks for the comment. The text has been corrected and now it is referred to a limitation described in the Spanish legislation. In any case, most of the Spanish legislation, especially that related to the protection of the environment, is translated from the European Directives, so usually the environmental parameters to meet are the same for all the EU Countries.

Line 56-58: the affermation is important, please, insert some other references, the only n.13 is not enough.

Thanks for the suggestion. In the following paragraph we have mentioned other studies that also treat wastewater, to complement the information. We have also included the following reference in the text:

Bernal, S., Drummond, J., Castelar, S., Gacia, E., Ribot, M., & Martí, E. (2020). Wastewater treatment plant effluent inputs induce large biogeochemical changes during low flows in an intermittent stream but small changes in day-night patterns. Science of the Total Environment, 714, 136733.

Line 84-85: please, give an explanation for which "wastewater is an oligotrophic environmnet".

Thanks for the comment. Urban wastewater is rich in nutrients but low in organic carbon compared to other wastewater sources (e.g., industrial wastewater), which does not favor the growth of PPB. For this reason, LF is included in the treatment to increase both contents and to obtain an optimal medium for the growth of PPB in relation to the ratio of COD:N:P. We have added the following text in the revised manuscript:

“In natural wastewater, which is an oligotrophic environment with low dissolved organic carbon content, …”

Line 97: SRT is conventionally "SLUDGE retention time", although it can be used generally for cells.

Thanks for the comment. The text has been corrected accordingly.

Line 127: why do not couple TSS to VSS?

Thanks for the comment. We have selected the most important wastewater parameters to simplify the information and ease the reading. The concentration of TSS and VSS is very similar as there are hardly any fixed suspended solids in the WW and LF, and it was removed from the graph to not overload the information.

Table 1: this is only a facultative suggestion: it could be preferable to use "sCOD" and "COD" rather than "SCOD" and "TCOD"; more, there is a * (star)  after the measure unit of VSS without an explanation.

Thanks for the suggestion. In the previous papers published by our team, we always used the acronyms TCOD and SCOD to refer to total and soluble COD. We would like to maintain it for clarity. The clarification for the (*) in Table 1 has been added at the end of the table.

Line 297: there is a typing mistake, "Figure 1" should be replaced by "Figure 2".

Thanks for the comment. It has been corrected.

References: please, prefer international references than national when it is possible, and verify and update the access of links.

Thanks for the suggestion. We have substituted national references for international ones when possible. We would like to maintain the reference to the Spanish legislation as it is the reference to what the experimental part of the paper has been designed. Also, we have checked and updated the references list to include all the correct links to access the documents.

Round 2

Reviewer 1 Report

Accept in present form

Reviewer 2 Report

All the requests of correction or clarification have been satisfied